# CLASSIFICATION-BASED ANOMALY DETECTION FOR GENERAL DATA

**Liron Bergman**   **Yedid Hoshen**
School of Computer Science and Engineering
The Hebrew University of Jerusalem, Israel

## ABSTRACT

Anomaly detection, finding patterns that substantially deviate from those seen previously, is one of the fundamental problems of artificial intelligence. Recently, classification-based methods were shown to achieve superior results on this task. In this work, we present a unifying view and propose an open-set method, GOAD, to relax current generalization assumptions. Furthermore, we extend the applicability of transformation-based methods to non-image data using random affine transformations. Our method is shown to obtain state-of-the-art accuracy and is applicable to broad data types. The strong performance of our method is extensively validated on multiple datasets from different domains.

## 1 INTRODUCTION

Detecting anomalies in perceived data is a key ability for humans and for artificial intelligence. Humans often detect anomalies to give early indications of danger or to discover unique opportunities. Anomaly detection systems are being used by artificial intelligence to discover credit card fraud, for detecting cyber intrusion, alert predictive maintenance of industrial equipment and for discovering attractive stock market opportunities. The typical anomaly detection setting is a one class classification task, where the objective is to classify data as normal or anomalous. The importance of the task stems from being able to raise an alarm when detecting a different pattern from those seen in the past, therefore triggering further inspection. This is fundamentally different from supervised learning tasks, in which examples of all data classes are observed.

There are different possible scenarios for anomaly detection methods. In supervised anomaly detection, we are given training examples of normal and anomalous patterns. This scenario can be quite well specified, however obtaining such supervision may not be possible. For example in cyber security settings, we will not have supervised examples of new, unknown computer viruses making supervised training difficult. On the other extreme, fully unsupervised anomaly detection, obtains a stream of data containing normal and anomalous patterns and attempts to detect the anomalous data. In this work we deal with the semi-supervised scenario. In this setting, we have a training set of normal examples (which contains no anomalies). After training the anomaly detector, we detect anomalies in the test data, containing both normal and anomalous examples. This supervision is easy to obtain in many practical settings and is less difficult than the fully-unsupervised case.

Many anomaly detection methods have been proposed over the last few decades. They can be broadly classified into reconstruction and statistically based methods. Recently, deep learning methods based on classification have achieved superior results. Most semi-supervised classification-based methods attempt to solve anomaly detection directly, despite only having normal training data. One example is: Deep-SVDD (Ruff et al., 2018) - one-class classification using a learned deep space. Another type of classification-based methods is self-supervised i.e. methods that solve one or more classification-based auxiliary tasks on the normal training data, and this is shown to be useful for solving anomaly detection, the task of interest e.g. (Golan & El-Yaniv, 2018). Self-supervised classification-based methods have been proposed with the object of image anomaly detection, but we show that by generalizing the class of transformations they can apply to all data types.

In this paper, we introduce a novel technique, GOAD, for anomaly detection which unifies current state-of-the-art methods that use normal training data only and are based on classification. Our method first transforms the data into $M$ subspaces, and learns a feature space such that inter-class

separation is larger than intra-class separation. For the learned features, the distance from the cluster center is correlated with the likelihood of anomaly. We use this criterion to determine if a new data point is normal or anomalous. We also generalize the class of transformation functions to include affine transformation which allows our method to generalize to non-image data. This is significant as tabular data is probably the most important for applications of anomaly detection. Our method is evaluated on anomaly detection on image and tabular datasets (cyber security and medical) and is shown to significantly improve over the state-of-the-art.

## 1.1 PREVIOUS WORKS

Anomaly detection methods can be generally divided into the following categories:

*Reconstruction Methods:* Some of the most common anomaly detection methods are reconstruction-based. The general idea behind such methods is that every normal sample should be reconstructed accurately using a limited set of basis functions, whereas anomalous data should suffer from larger reconstruction costs. The choice of features, basis and loss functions differentiates between the different methods. Some of the earliest methods use: nearest neighbors (Eskin et al., 2002), low-rank PCA (Jolliffe, 2011; Candès et al., 2011) or K-means (Hartigan & Wong, 1979) as the reconstruction basis. Most recently, neural networks were used (Sakurada & Yairi, 2014; Xia et al., 2015) for learning deep basis functions for reconstruction. Another set of recent methods (Schlegl et al., 2017; Deecke et al., 2018) use GANs to learn a reconstruction basis function. GANs suffer from mode-collapse and are difficult to invert, which limits the performance of such methods.

*Distributional Methods:* Another set of commonly used methods are distribution-based. The main theme in such methods is to model the distribution of normal data. The expectation is that anomalous test data will have low likelihood under the probabilistic model while normal data will have higher likelihoods. Methods differ in the features used to describe the data and the probabilistic model used to estimate the normal distribution. Some early methods used Gaussian or Gaussian mixture models. Such models will only work if the data under the selected feature space satisfies the probabilistic assumptions implicied by the model. Another set of methods used non-parametric density estimate methods such as kernel density estimate (Parzen, 1962). Recently, deep learning methods (autoencoders or variational autoencoders) were used to learn deep features which are sometimes easier to model than raw features (Yang et al., 2017). DAGMM introduced by Zong et al. (2018) learn the probabilistic model jointly with the deep features therefore shaping the features space to better conform with the probabilistic assumption.

*Classification-Based Methods:* Another paradigm for anomaly detection is separation between space regions containing normal data from all other regions. An example of such approach is One-Class SVM (Scholkopf et al., 2000), which trains a classifier to perform this separation. Learning a good feature space for performing such separation is performed both by the classic kernel methods as well as by the recent deep learning approach (Ruff et al., 2018). One of the main challenges in unsupervised (or semi-supervised) learning is providing an objective for learning features that are relevant to the task of interest. One method for learning good representations in a self-supervised way is by training a neural network to solve an auxiliary task for which obtaining data is free or at least very inexpensive. Auxiliary tasks for learning high-quality image features include: video frame prediction (Mathieu et al., 2016), image colorization (Zhang et al., 2016; Larsson et al., 2016), puzzle solving (Noroozi & Favaro, 2016) - predicting the correct order of random permuted image patches. Recently, Gidaris et al. (2018) used a set of image processing transformations (rotation by $0, 90, 180, 270$ degrees around the image axis, and predicted the true image orientation has been used to learn high-quality image features. Golan & El-Yaniv (2018), have used similar image-processing task prediction for detecting anomalies in images. This method has shown good performance on detecting images from anomalous classes. In this work, we overcome some of the limitations of previous classification-based methods and extend their applicability of self-supervised methods to general data types. We also show that our method is more robust to adversarial attacks.

## 2 CLASSIFICATION-BASED ANOMALY DETECTION

Classification-based methods have dominated supervised anomaly detection. In this section we will analyse semi-supervised classification-based methods:

Let us assume all data lies in space $R^L$ (where $L$ is the data dimension). Normal data lie in subspace $X \subset R^L$. We assume that all anomalies lie outside $X$. To detect anomalies, we would therefore like to build a classifier $C$, such that $C(x) = 1$ if $x \in X$ and $C(x) = 0$ if $x \in R^L \backslash X$.

One-class classification methods attempt to learn $C$ directly as $P(x \in X)$. Classical approaches have learned a classifier either in input space or in a kernel space. Recently, Deep-SVDD (Ruff et al., 2018) learned end-to-end to i) transform the data to an isotropic feature space $f(x)$ ii) fit the minimal hypersphere of radius $R$ and center $c_0$ around the features of the normal training data. Test data is classified as anomalous if the following normality score is positive: $\|f(x) - c_0\|^2 - R^2$. Learning an effective feature space is not a simple task, as the trivial solution of $f(x) = 0 \; \forall \; x$ results in the smallest hypersphere, various tricks are used to avoid this possibility.

Geometric-transformation classification (GEOM), proposed by Golan & El-Yaniv (2018) first transforms the normal data subspace $X$ into $M$ subspaces $X_1..X_M$. This is done by transforming each image $x \in X$ using $M$ different geometric transformations (rotation, reflection, translation) into $T(x, 1)..T(x, M)$. Although these transformations are image specific, we will later extend the class of transformations to all affine transformations making this applicable to non-image data. They set an auxiliary task of learning a classifier able to predict the transformation label $m$ given transformed data point $T(x, m)$. As the training set consists of normal data only, each sample is $x \in X$ and the transformed sample is in $\cup_m X_m$. The method attempts to estimate the following conditional probability:

$$P(m'|T(x,m)) = \frac{P(T(x,m) \in X_{m'})P(m')}{\sum_{\tilde{m}} P(T(x,m) \in X_{\tilde{m}})P(\tilde{m})} = \frac{P(T(x,m) \in X_{m'})}{\sum_{\tilde{m}} P(T(x,m) \in X_{\tilde{m}})} \tag{1}$$

Where the second equality follows by design of the training set, and where every training sample is transformed exactly once by each transformation leading to equal priors.

For anomalous data $x \in R^L \backslash X$, by construction of the subspace, if the transformations $T$ are one-to-one, it follows that the transformed sample does not fall in the appropriate subspace: $T(x, m) \in R^L \backslash X_m$. GEOM uses $P(m|T(x,m))$ as a score for determining if $x$ is anomalous i.e. that $x \in R^L \backslash X$. GEOM gives samples with low probabilities $P(m|T(x,m))$ high anomaly scores.

A significant issue with this methodology, is that the learned classifier $P(m'|T(x,m))$ is only valid for samples $x \in X$ which were found in the training set. For $x \in R^L \backslash X$ we should in fact have $P(T(x,m) \in X_{m'}) = 0$ for all $m = 1..M$ (as the transformed $x$ is not in any of the subsets). This makes the anomaly score $P(m'|T(x,m))$ have very high variance for anomalies.

One way to overcome this issue is by using examples of anomalies $x_a$ and training $P(m|T(x,m)) = \frac{1}{M}$ on anomalous data. This corresponds to the supervised scenario and was recently introduced as Outlier Exposure (Hendrycks et al., 2018). Although getting such supervision is possible for some image tasks (where large external datasets can be used) this is not possible in the general case e.g. for tabular data which exhibits much more variation between datasets.

## 3 DISTANCE-BASED MULTIPLE TRANSFORMATION CLASSIFICATION

We propose a novel method to overcome the generalization issues highlighted in the previous section by using ideas from open-set classification (Bendale & Boult, 2016). Our approach unifies one-class and transformation-based classification methods. Similarly to GEOM, we transform $X$ to $X_1..X_M$. We learn a feature extractor $f(x)$ using a neural network, which maps the original input data into a feature representation. Similarly to deep OC methods, we model each subspace $X_m$ mapped to the feature space $\{f(x)|x \in X_m\}$ as a sphere with center $c_m$. The probability of data point $x$ after transformation $m$ is parameterized by $P(T(x,m) \in X'_m) = \frac{1}{Z}e^{-(f(T(x,m))-c'_m)^2}$. The classifier predicting transformation $m$ given a transformed point is therefore:

$$P(m'|T(x,m)) = \frac{e^{-\|f(T(x,m))-c_{m'}\|^2}}{\sum_{\tilde{m}} e^{-\|f(T(x,m))-c_{\tilde{m}}\|^2}} \tag{2}$$

The centers $c_m$ are given by the average feature over the training set for every transformation i.e. $c_m = \frac{1}{N}\sum_{x \in X} f(T(x,m))$. One option is to directly learn $f$ by optimizing cross-entropy between

$P(m'|T(x,m))$ and the correct label on the normal training set. In practice we obtained better results by training $f$ using the center triplet loss (He et al., 2018), which learns supervised clusters with low intra-class variation, and high-inter-class variation by optimizing the following loss function (where $s$ is a margin regularizing the distance between clusters):

$$L = \sum_i \max(\|f(T(x_i, m)) - c_m\|^2 + s - min_{m' \neq m}\|f(T(x_i, m)) - c_{m'}\|^2, 0) \qquad (3)$$

Having learned a feature space in which the different transformation subspaces are well separated, we use the probability in Eq. 2 as a normality score. However, for data far away from the normal distributions, the distances from the means will be large. A small difference in distance will make the classifier unreasonably certain of a particular transformation. To add a general prior for uncertainty far from the training set, we add a small regularizing constant $\epsilon$ to the probability of each transformation. This ensures equal probabilities for uncertain regions:

$$\tilde{P}(m'|T(x,m)) = \frac{e^{-\|f(T(x,m))-c_{m'}\|^2} + \epsilon}{\sum_{\tilde{m}} e^{-\|f(T(x,m))-c_{\tilde{m}}\|^2} + M \cdot \epsilon} \qquad (4)$$

At test time we transform each sample by the $M$ transformations. By assuming independence between transformations, the probability that $x$ is normal (i.e. $x \in X$) is the product of the probabilities that all transformed samples are in their respective subspace. For log-probabilities the total score is given by:

$$Score(x) = -\log P(x \in X) = -\sum_m \log \tilde{P}(T(x,m) \in X_m) = -\sum_m \log \tilde{P}(m|T(x,m)) \quad (5)$$

The score computes the degree of anomaly of each sample. Higher scores indicate a more anomalous sample.

---
**Algorithm 1** GOAD: Training Algorithm
---
**Input:** Normal training data $x_1, x_2...x_N$
      Transformations $T(, 1), T(, 2)...T(, M)$
**Output:** Feature extractor $f$, centers $c_1, c_2...c_M$
  $T(x_i, 1), T(x_i, 2)...T(x_i, M) \leftarrow x_i$
    // Transform each sample by all transformations 1 to $M$
  Find $f, c_1, c_2...c_M$ that optimize the triplet loss in Eq. 3

---

---
**Algorithm 2** GOAD: Evaluation Algorithm
---
**Input:** Test sample: $x$, feature extractor: $f$, centers: $c_1, c_2...c_M$, transformations: $T(, 1), T(, 2)...T(, M)$
**Output:** Score(x)
  $T(x, 1), T(x, 2)...T(x, M) \leftarrow x$
    // Transform test sample by all transformations 1 to $M$
  $P(m|T(x,m)) \leftarrow f(T(x,m)), c_1, c_2...c_M$
    // Likelihood of predicting the correct transformation (Eq. 4)
  $Score(x) \leftarrow P(1|T(x,1)), P(2|T(x,2))...P(M|T(x,M))$
    // Aggregate probabilities to compute anomaly score (Eq. 5)

---

## 4 PARAMETERIZING THE SET OF TRANSFORMATIONS

Geometric transformations have been used previously for unsupervised feature learning by Gidaris et al. (2018) as well as by GEOM (Golan & El-Yaniv, 2018) for classification-based anomaly detection. This set of transformations is hand-crafted to work well with convolutional neural networks

(CNNs) which greatly benefit from preserving neighborhood between pixels. This is however not a requirement for fully-connected networks.

Anomaly detection often deals with non-image datasets e.g. tabular data. Tabular data is very commonly used on the internet e.g. for cyber security or online advertising. Such data consists of both discrete and continuous attributes with no particular neighborhoods or order. The data is one-dimensional and rotations do not naturally generalize to it. To allow transformation-based methods to work on general data types, we therefore need to extend the class of transformations.

We propose to generalize the set of transformations to the class of affine transformations (where we have a total of $M$ transformations):

$$T(x, m) = W_m x + b_m \tag{6}$$

It is easy to verify that all geometric transformations in Golan & El-Yaniv (2018) (rotation by a multiple of 90 degrees, flips and translations) are a special case of this class ($x$ in this case is the set of image pixels written as a vector). The affine class is however much more general than mere permutations, and allows for dimensionality reduction, non-distance preservation and random transformation by sampling $W$, $b$ from a random distribution.

Apart from reduced variance across different dataset types where no apriori knowledge on the correct transformation classes exists, random transformations are important for avoiding adversarial examples. Assume an adversary wishes to change the label of a particular sample from anomalous to normal or vice versa. This is the same as requiring that $\tilde{P}(m'|T(x, m))$ has low or high probability for $m' = m$. If $T$ is chosen deterministically, the adversary may create adversarial examples against the known class of transformations (even if the exact network parameters are unknown). Conversely, if $T$ is unknown, the adversary must create adversarial examples that generalize across different transformations, which reduces the effectiveness of the attack.

To summarize, generalizing the set of transformations to the affine class allows us to: generalize to non-image data, use an unlimited number of transformations and choose transformations randomly which reduces variance and defends against adversarial examples.

## 5 EXPERIMENTS

We perform experiments to validate the effectiveness of our distance-based approach and the performance of the general class of transformations we introduced for non-image data.

### 5.1 IMAGE EXPERIMENTS

**Cifar10:** To evaluate the performance of our method, we perform experiments on the Cifar10 dataset. We use the same architecture and parameter choices of Golan & El-Yaniv (2018), with our distance-based approach. We use the standard protocol of training on all training images of a single digit and testing on all test images. Results are reported in terms of AUC. In our method, we used a margin of $s = 0.1$ (we also run GOAD with $s = 1$, shown in the appendix). Similarly to He et al. (2018), to stabilize training, we added a softmax + cross entropy loss, as well as $L_2$ norm regularization for the extracted features $f(x)$. We compare our method with the deep one-class method of Ruff et al. (2018) as well as Golan & El-Yaniv (2018) without and with Dirichlet weighting. We believe the correct comparison is without Dirichlet post-processing, as we also do not use it in our method. Our distance based approach outperforms the SOTA approach by Golan & El-Yaniv (2018), both with and without Dirichlet (which seems to improve performance on a few classes). This gives evidence for the importance of considering the generalization behavior outside the normal region used in training. Note that we used the same geometric transformations as Golan & El-Yaniv (2018). Random affine matrices did not perform competitively as they are not pixel order preserving, this information is effectively used by CNNs and removing this information hurts performance. This is a special property of CNN architectures and image/time series data. As a rule of thumb, fully-connected networks are not pixel order preserving and can fully utilize random affine matrices.

Table 1: Anomaly Detection Accuracy on Cifar10 (ROC-AUC %)

| Class | Method | | | |
|---|---|---|---|---|
| | Deep-SVDD | GEOM (no Dirichlet) | GEOM (w. Dirichlet) | Ours |
| 0 | $61.7 \pm 1.3$ | $76.0 \pm 0.8$ | $74.7 \pm 0.4$ | $\mathbf{77.2} \pm 0.6$ |
| 1 | $65.9 \pm 0.7$ | $83.0 \pm 1.6$ | $95.7 \pm 0.0$ | $\mathbf{96.7} \pm 0.2$ |
| 2 | $50.8 \pm 0.3$ | $79.5 \pm 0.7$ | $78.1 \pm 0.4$ | $\mathbf{83.3} \pm 1.4$ |
| 3 | $59.1 \pm 0.4$ | $71.4 \pm 0.9$ | $72.4 \pm 0.5$ | $\mathbf{77.7} \pm 0.7$ |
| 4 | $60.9 \pm 0.3$ | $83.5 \pm 1.0$ | $\mathbf{87.8} \pm 0.2$ | $\mathbf{87.8} \pm 0.7$ |
| 5 | $65.7 \pm 0.8$ | $84.0 \pm 0.3$ | $\mathbf{87.8} \pm 0.1$ | $\mathbf{87.8} \pm 0.6$ |
| 6 | $67.7 \pm 0.8$ | $78.4 \pm 0.7$ | $83.4 \pm 0.5$ | $\mathbf{90.0} \pm 0.6$ |
| 7 | $67.3 \pm 0.3$ | $89.3 \pm 0.5$ | $95.5 \pm 0.1$ | $\mathbf{96.1} \pm 0.3$ |
| 8 | $75.9 \pm 0.4$ | $88.6 \pm 0.6$ | $93.3 \pm 0.0$ | $\mathbf{93.8} \pm 0.9$ |
| 9 | $73.1 \pm 0.4$ | $82.4 \pm 0.7$ | $91.3 \pm 0.1$ | $\mathbf{92.0} \pm 0.6$ |
| Average | 64.8 | 81.6 | 86.0 | **88.2** |

Table 2: Anomaly Detection Accuracy on FashionMNIST (ROC-AUC %)

| Class | Method | | | |
|---|---|---|---|---|
| | Deep-SVDD | GEOM (no Dirichlet) | GEOM (w. Dirichlet) | Ours |
| 0 | 98.2 | $77.8 \pm 5.9$ | $\mathbf{99.4} \pm 0.0$ | $94.1 \pm 0.9$ |
| 1 | 90.3 | $79.1 \pm 16.3$ | $97.6 \pm 0.1$ | $\mathbf{98.5} \pm 0.3$ |
| 2 | 90.7 | $80.8 \pm 6.9$ | $\mathbf{91.1} \pm 0.2$ | $90.8 \pm 0.4$ |
| 3 | 94.2 | $79.2 \pm 9.1$ | $89.9 \pm 0.4$ | $\mathbf{91.6} \pm 0.9$ |
| 4 | 89.4 | $77.8 \pm 3.3$ | $\mathbf{92.1} \pm 0.0$ | $91.4 \pm 0.3$ |
| 5 | 91.8 | $58.0 \pm 29.4$ | $93.4 \pm 0.9$ | $\mathbf{94.8} \pm 0.5$ |
| 6 | 83.4 | $73.6 \pm 8.7$ | $83.3 \pm 0.1$ | $\mathbf{83.4} \pm 0.4$ |
| 7 | 98.8 | $87.4 \pm 11.4$ | $\mathbf{98.9} \pm 0.1$ | $97.9 \pm 0.4$ |
| 8 | 91.9 | $84.6 \pm 5.6$ | $90.8 \pm 0.1$ | $\mathbf{98.9} \pm 0.1$ |
| 9 | 99.0 | $\mathbf{99.5} \pm 0.0$ | $99.2 \pm 0.0$ | $99.2 \pm 0.3$ |
| Average | 92.8 | 79.8 | 93.5 | **94.1** |

**FasionMNIST:** In Tab. 2, we present a comparison between our method (GOAD) and the strongest baseline methods (Deep SVDD and GEOM) on the FashionMNIST dataset. We used exactly the same setting as Golan & El-Yaniv (2018). GOAD was run with $s = 1$. OCSVM and GEOM with Dirichlet were copied from their paper. We run their method without Dirichlet and presented it in the table (we verified the implementation by running their code with Dirichlet and replicated the numbers in the paper). It appears that GEOM is quite dependent on Dirichlet for this dataset, whereas we do not use it at all. GOAD outperforms all the baseline methods.

**Adversarial Robustness:** Let us assume an attack model where the attacker knows the architecture and the normal training data and is trying to minimally modify anomalies to look normal. We examine the merits of two settings i) the adversary knows the transformations used (non-random) ii) the adversary uses another set of transformations. To measure the benefit of the randomized transformations, we train three networks A, B, C. Networks A and B use exactly the same transformations but random parameter initialization prior to training. Network C is trained using other randomly selected transformations. The adversary creates adversarial examples using PGD (Madry et al., 2017) based on network A (making anomalies appear like normal data). On Cifar10, we randomly selected 8 transformations from the full set of 72 for $A$ and $B$, another randomly selected 8 transformations are used for $C$. We measure the increase of false classification rate on the adversarial examples using the three networks. The average increase in performance of classifying transformation correctly on anomalies (causing lower anomaly scores) on the original network $A$ was 12.8%, the transfer performance for B causes an increase by 5.0% on network $B$ which shared the same set of transformation, and 3% on network $C$ that used other rotations. This shows the benefits of using random transformations.

Table 3: Anomaly Detection Accuracy (%)

| Method | Dataset | | | | | | | |
|---|---|---|---|---|---|---|---|---|
| | Arrhythmia | | Thyroid | | KDD | | KDDRev | |
| | $F_1$ Score | $\sigma$ | $F_1$ Score | $\sigma$ | $F_1$ Score | $\sigma$ | $F_1$ Score | $\sigma$ |
| OC-SVM | 45.8 | | 38.9 | | 79.5 | | 83.2 | |
| E2E-AE | 45.9 | | 11.8 | | 0.3 | | 74.5 | |
| LOF | 50.0 | 0.0 | 52.7 | 0.0 | 83.8 | 5.2 | 81.6 | 3.6 |
| DAGMM | 49.8 | | 47.8 | | 93.7 | | 93.8 | |
| FB-AE | **51.5** | 1.6 | **75.0** | 0.8 | 92.7 | 0.3 | 95.9 | 0.4 |
| GOAD(Ours) | **52.0** | 2.3 | **74.5** | 1.1 | **98.4** | 0.2 | **98.9** | 0.3 |

## 5.2 TABULAR DATA EXPERIMENTS

*Datasets:* We evaluate on small-scale medical datasets Arrhythmia, Thyroid as well as large-scale cyber intrusion detection datasets KDD and KDDRev. Our configuration follows that of Zong et al. (2018). Categorical attributes are encoded as one-hot vectors. For completeness the datasets are described in the appendix A.2. We train all compared methods on $50\%$ of the normal data. The methods are evaluated on $50\%$ of the normal data as well as all the anomalies.

*Baseline methods:* The baseline methods evaluated are: One-Class SVM (OC-SVM, Scholkopf et al. (2000)), End-to-End Autoencoder (E2E-AE), Local Outlier Factor (LOF, Breunig et al. (2000)). We also evaluated deep distributional method DAGMM (Zong et al., 2018), choosing their strongest variant. To compare against ensemble methods e.g. Chen et al. (2017), we implemented the Feature Bagging Autoencoder (FB-AE) with autoencoders as the base classifier, feature bagging as the source of randomization, and average reconstruction error as the anomaly score. OC-SVM, E2E-AE and DAGMM results are directly taken from those reported by Zong et al. (2018). LOF and FB-AE were computed by us.

*Implementation of GOAD:* We randomly sampled transformation matrices using the normal distribution for each element. Each matrix has dimensionality $L \times r$, where $L$ is the data dimension and $r$ is a reduced dimension. For Arryhthmia and Thyroid we used $r = 32$, for KDD and KDDrev we used $r = 128$ and $r = 64$ respectively, the latter due to high memory requirements. We used 256 tasks for all datasets apart from KDD (64) due to high memory requirements. We set the bias term to 0. For $C$ we used fully-connected hidden layers and leaky-ReLU activations (8 hidden nodes for the small datasets, 128 and 32 for KDDRev and KDD). We optimized using ADAM with a learning rate of 0.001. Similarly to He et al. (2018), to stabilize the triplet center loss training, we added a softmax + cross entropy loss. We repeated the large-scale experiments 5 times, and the small scale GOAD experiments 500 times (due to the high variance). We report the mean and standard deviation ($\sigma$). Following the protocol in Zong et al. (2018), the decision threshold value is chosen to result in the correct number of anomalies e.g. if the test set contains $N_a$ anomalies, the threshold is selected so that the highest $N_a$ scoring examples are classified as anomalies. True positives and negatives are evaluated in the usual way. Some experiments copied from other papers did not measure standard variation and we kept the relevant cell blank.

**Results**

*Arrhythmia:* The Arrhythmia dataset was the smallest examined. A quantitative comparison on this dataset can be seen in Tab. 3. OC-SVM and DAGMM performed reasonably well. Our method is comparable to FB-AE. A linear classifier $C$ performed better than deeper networks (which suffered from overfitting). Early stopping after a single epoch generated the best results.

*Thyroid:* Thyroid is a small dataset, with a low anomaly to normal ratio and low feature dimensionality. A quantitative comparison on this dataset can be seen in Tab. 3. Most baselines performed about equally well, probably due to the low dimensionality. On this dataset, we also found that early stopping after a single epoch gave the best results. The best results on this dataset, were obtained with a linear classifier. Our method is comparable to FB-AE and beat all other baselines by a wide margin.

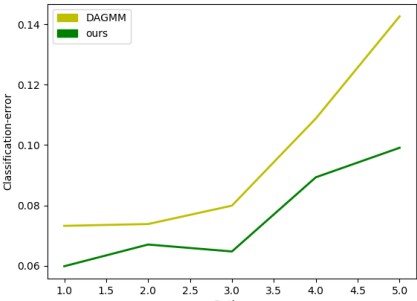 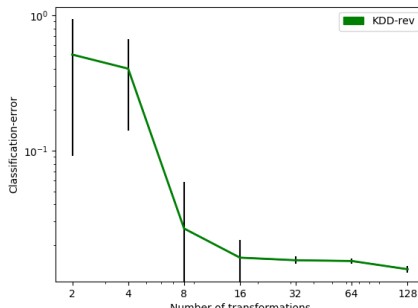

Figure 1: Left: Classification error for our method and DAGMM as a function of percentage of the anomalous examples in the training set (on the KDDCUP99 dataset). Our method consistently outperforms the baseline. Right: Classification error as a function of the number of transformations (on the KDDRev dataset). The error and instability decrease as a function of the number of transformations. For both, lower is better.

*KDDCUP99:* The UCI KDD $10\%$ dataset is the largest dataset examined. A quantitative comparison on this dataset can be seen in Tab. 3. The strongest baselines are FB-AE and DAGMM. Our method significantly outperformed all baselines. We found that large datasets have different dynamics from very small datasets. On this dataset, deep networks performed the best. We also, did not need early stopping. The results are reported after 25 epochs.

*KDD-Rev:* The KDD-Rev dataset is a large dataset, but smaller than KDDCUP99 dataset. A quantitative comparison on this dataset can be seen in Tab. 3. Similarly to KDDCUP99, the best baselines are FB-AE and DAGMM, where FB-AE significantly outperforms DAGMM. Our method significantly outperformed all baselines. Due to the size of the dataset, we did not need early stopping. The results are reported after 25 epochs.

*Adversarial Robustness:* Due to the large number of transformations and relatively small networks, adversarial examples are less of a problem for tabular data. PGD generally failed to obtain adversarial examples on these datasets. On KDD, transformation classification accuracy on anomalies was increased by $3.7\%$ for the network the adversarial examples were trained on, $1.3\%$ when transferring to the network with the same transformation and only $0.2\%$ on the network with other randomly selected transformations. This again shows increased adversarial robustness due to random transformations.

**Further Analysis**

*Contaminated Data:* This paper deals with the semi-supervised scenario i.e. when the training dataset contains only normal data. In some scenarios, such data might not be available but instead we might have a training dataset that contains a small percentage of anomalies. To evaluate the robustness of our method to this unsupervised scenario, we analysed the KDDCUP99 dataset, when $X\%$ of the training data is anomalous. To prepare the data, we used the same normal training data as before and added further anomalous examples. The test data consists of the same proportions as before. The results are shown in Fig. 1. Our method significantly outperforms DAGMM for all impurity values, and degrades more graceful than the baseline. This attests to the effectiveness of our approach. Results for the other datasets are presented in Fig. 3, showing similar robustness to contamination.

*Number of Tasks:* One of the advantages of GOAD, is the ability to generate any number of tasks. We present the anomaly detection performance on the KDD-Rev dataset with different numbers of tasks in Fig. 1. We note that a small number of tasks (less than 16) leads to poor results. From 16 tasks, the accuracy remains stable. We found that on the smaller datasets (Thyroid, Arrhythmia) using a larger number of transformations continued to reduce $F_1$ score variance between differently initialized runs (Fig. 2).

## 6 DISCUSSION

*Openset vs. Softmax:* The openset-based classification presented by GOAD resulted in performance improvement over the closed-set softmax approach on Cifar10 and FasionMNIST. In our experiments, it has also improved performance in KDDRev. Arrhythmia and Thyroid were comparable. As a negative result, performance of softmax was better on KDD ($F_1 = 0.99$).

*Choosing the margin parameter $s$:* GOAD is not particularly sensitive to the choice of margin parameter $s$, although choosing $s$ that is too small might cause some instability. We used a fixed value of $s = 1$ in our experiments, and recommend this value as a starting point.

*Other transformations:* GOAD can also work with other types of transformations such as rotations or permutations for tabular data. In our experiments, we observed that these transformation types perform comparably but a little worse than affine transformations.

*Unsupervised training:* Although most of our results are semi-supervised i.e. assume that no anomalies exist in the training set, we presented results showing that our method is more robust than strong baselines to a small percentage of anomalies in the training set. We further presented results in other datasets showing that our method degrades gracefully with a small amount of contamination. Our method might therefore be considered in the unsupervised settings.

*Deep vs. shallow classifiers:* Our experiments show that for large datasets deep networks are beneficial (particularly for the full KDDCUP99), but are not needed for smaller datasets (indicating that deep learning has not benefited the smaller datasets). For performance critical operations, our approach may be used in a linear setting. This may also aid future theoretical analysis of our method.

## 7 CONCLUSION

In this paper, we presented a method for detecting anomalies for general data. This was achieved by training a classifier on a set of random auxiliary tasks. Our method does not require knowledge of the data domain, and we are able to generate an arbitrary number of random tasks. Our method significantly improve over the state-of-the-art.

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

## A  APPENDIX

### A.1  IMAGE EXPERIMENTS

**Sensitive to margin** $s$**:** We run Cifar10 experiments with $s = 0.1$ and $s = 1$ and presented the results in Fig. 4. The results were not affected much by the margin parameter. This is in-line with the rest of our empirical observations that GOAD is not very sensitive to the margin parameter.

Table 4: Anomaly Detection Accuracy on Cifar10 (%)

| Class | Method | | |
|---|---|---|---|
| | GEOM (w. Dirichlet) | GOAD($s = 0.1$) | GOAD(1.0) |
| 0 | $74.7 \pm 0.4$ | $77.2 \pm 0.6$ | $\mathbf{77.9} \pm 0.7$ |
| 1 | $95.7 \pm 0.0$ | $\mathbf{96.7} \pm 0.2$ | $96.4 \pm 0.9$ |
| 2 | $78.1 \pm 0.4$ | $\mathbf{83.3} \pm 1.4$ | $81.8 \pm 0.8$ |
| 3 | $72.4 \pm 0.5$ | $\mathbf{77.7} \pm 0.7$ | $77.0 \pm 0.7$ |
| 4 | $\mathbf{87.8} \pm 0.2$ | $\mathbf{87.8} \pm 0.7$ | $87.7 \pm 0.5$ |
| 5 | $\mathbf{87.8} \pm 0.1$ | $\mathbf{87.8} \pm 0.6$ | $\mathbf{87.8} \pm 0.7$ |
| 6 | $83.4 \pm 0.5$ | $90.0 \pm 0.6$ | $\mathbf{90.9} \pm 0.5$ |
| 7 | $95.5 \pm 0.1$ | $\mathbf{96.1} \pm 0.3$ | $96.1 \pm 0.2$ |
| 8 | $93.3 \pm 0.0$ | $\mathbf{93.8} \pm 0.9$ | $93.3 \pm 0.1$ |
| 9 | $91.3 \pm 0.1$ | $92.0 \pm 0.6$ | $\mathbf{92.4} \pm 0.3$ |
| Average | 86.0 | **88.2** | 88.1 |

## A.2 TABULAR DATASETS

Following the evaluation protocol of Zong et al. (2018), 4 datasets are used in this comparison:

*Arrhythmia:* A cardiology dataset from the UCI repository (Asuncion & Newman, 2007) containing attributes related to the diagnosis of cardiac arrhythmia in patients. The datasets consists of 16 classes: class 1 are normal patients, 2-15 contain different arrhythmia conditions, and class 16 contains undiagnosed cases. Following the protocol established by ODDS (Rayana, 2016), the smallest classes: $3, 4, 5, 7, 8, 9, 14, 15$ are taken to be anomalous and the rest normal. Also following ODDS, the categorical attributes are dropped, the final attributes total $274$.

*Thyroid:* A medical dataset from the UCI repository (Asuncion & Newman, 2007), containing attributes related to whether a patient is hyperthyroid. Following ODDS (Rayana, 2016), from the 3 classes of the dataset, we designate hyperfunction as the anomalous class and the rest as normal. Also following ODDS only the 6 continuous attributes are used.

*KDD:* The KDD Intrusion Detection dataset was created by an extensive simulation of a US Air Force LAN network. The dataset consists of the normal and 4 simulated attack types: denial of service, unauthorized access from a remote machine, unauthorized access from local superuser and probing. The dataset consists of around 5 million TCP connection records. Following the evaluation protocol in Zong et al. (2018), we use the UCI KDD $10\%$ dataset, which is a subsampled version of the original dataset. The dataset contains $41$ different attributes. $34$ are continuous and $7$ are categorical. Following Zong et al. (2018), we encode the categorical attributes using 1-hot encoding.

Following Zong et al. (2018), we evaluate two different settings for the KDD dataset:

*KDDCUP99:* In this configuration we use the entire UCI $10\%$ dataset. As the non-attack class consists of only $20\%$ of the dataset, it is treated as the anomaly in this case, while attacks are treated as normal.

*KDDCUP99-Rev:* To better correspond to the actual use-case, in which the non-attack scenario is normal and attacks are anomalous, Zong et al. (2018) also evaluate on the reverse configuration, in which the attack data is sub-sampled to consist of $25\%$ of the number of non-attack samples. The attack data is in this case designated as anomalous (the reverse of the KDDCUP99 dataset).

In all the above datasets, the methods are trained on $50\%$ of the normal data. The methods are evaluated on $50\%$ of the normal data as well as all the anomalies.

## A.3 NUMBER OF TASKS

We provide plots of the number of auxiliary tasks vs. the anomaly detection accuracy (measured by $F_1$) for all datasets. The results are presented in Fig. 2. Performance increases rapidly up to a certain number of tasks (around 16). Afterwards more tasks reduce the variance of $F_1$ scores between runs.

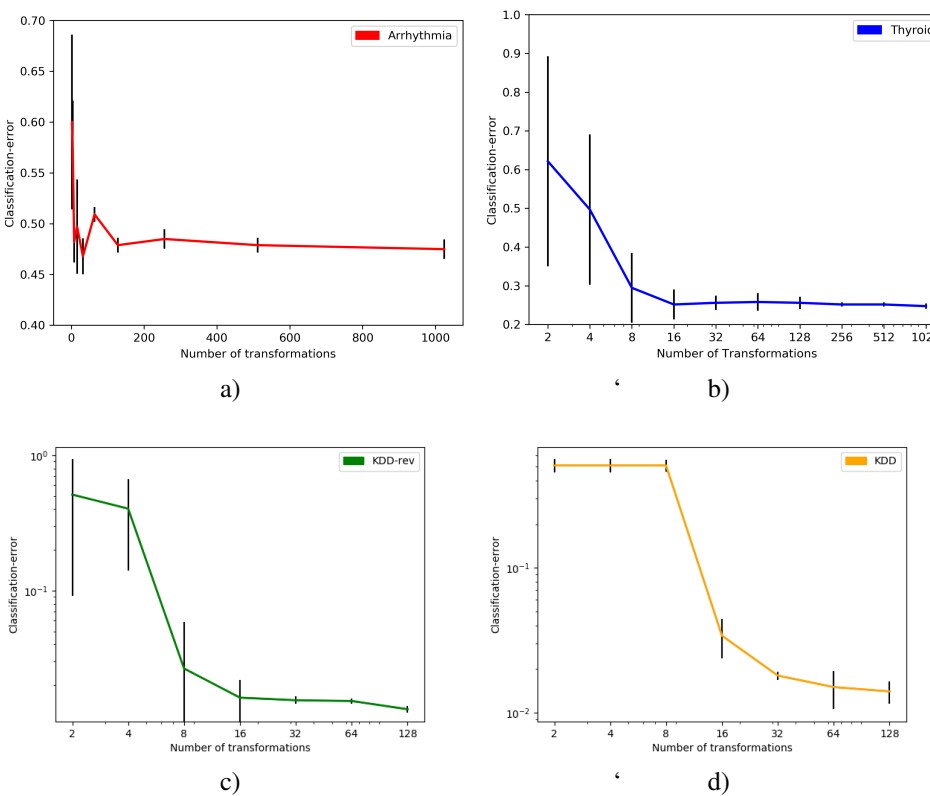

Figure 2: Plots of the number of auxiliary tasks vs. the anomaly detection accuracy (measured by $F_1$) a) Arrhythmia b) Thyroid c) KDDRev d) KDDCup99 Accuracy often increases with the number of tasks, although the rate diminishes with the number of tasks.

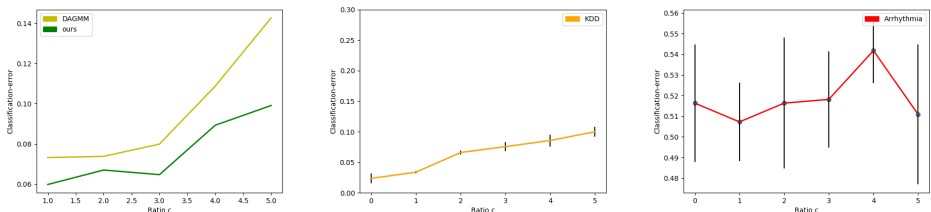

Figure 3: Plots of the degree of contamination vs. the anomaly detection accuracy (measured by $F_1$) (left) KDDRev (center) KDDCup99 (right) Arrhythmia. GOAD is generally robust to the degree of contamination.

## A.4 CONTAMINATION EXPERIMENTS

We conduct contamination experiments for 3 datasets. Thyroid was omitted due to not having a sufficient number of anomalies. The protocol is different than that of KDDRev as we do not have unused anomalies for contamination. Instead, we split the anomalies into train and test. Train anomalies are used for contamination, test anomalies are used for evaluation. As DAGMM did not present results for the other datasets, we only present GOAD. GOAD was reasonably robust to contamination on KDD, KDDRev and Arrhythmia. The results are presented in Fig. 3

