# OpenReview forum: "Classification-Based Anomaly Detection for General Data"
_ICLR.cc/2020/Conference — Accept (Poster)_

### Official Review · AnonReviewer3 · 2019-10-21
**Official Blind Review #3**

**Rating:** 6

**Review:**

UPDATE:
I acknowledge that I‘ve read the author responses as well as the other reviews.

I appreciate the clarifications, additional experiments, and overall improvements made to the paper. I updated my score to 6 Weak Accept.


####################

This paper proposes a deep method for anomaly detection (AD) that unifies recent deep one-class classification [6] and transformation-based classification [3, 4] approaches. The proposed method transforms the data to $M$ subspaces via $M$ random affine transformations and identifies with each such transformation a cluster centered around some centroid (set as the mean of the respectively transformed samples). The training objective of the method is defined by the triplet loss [5] which learns to separate the subspaces via maximizing the inter-class as well as minimizing the intra-class variation. The anomaly score for a sample is finally given by the sum of log-probabilities, where each transformation-/cluster-probability is derived from the distance to the cluster center. Using random affine transformations, the proposed method is applicable to general data types in contrast to previous works that only consider geometric transformations (rotation, translation, etc.) on image data [3, 4]. The paper conclusively presents experiments on CIFAR-10 and four tabular datasets (Arrhythmia, Thyroid, KDD, KDD-Rev) that indicate a superior detection performance of the proposed method over baselines and deep competitors.

I think this paper is not yet ready for acceptance due to the following main reason:
(i) The experimental evaluation needs clarification and should be extended to judge the significance of the empirical results.

(i) I think the comparison with state-of-the-art deep competitors [6, 4] should consider at least another image dataset besides CIFAR-10, e.g. Fashion-MNIST or the recently published MVTec [1] for AD. On CIFAR-10, do you also consider geometric transformations however using your triplet loss or are the reported results from random affine transformations? I think reporting both would be insightful to see the difference between image-specific and random affine transformations.
On the tabular datasets, how do deep networks perform in contrast to the final linear classifier reported on most datasets? Especially when only using a final linear classifier, the proposed method is very similar to ensemble learning on random subspace projections. Figure 1 (right) shows an error curve that is also typical for ensemble learning (decrease in mean error and reduction in overall variance). I think this should be discussed and ensemble baselines [2] should be considered for a fair comparison. Table 2 also seems incomplete with the variances missing for some methods?
Further clarifications are needed. How many transformations $M$ do you consider on the specific datasets? How is hyperparameter $s$ chosen?
Finally, I think the claim that the approach is robust against training data contamination is too early from only comparing against the DAGMM method on KDDCUP (Is Figure 1 (left) wrong labeled? As presented DAGMM shows a lower classification error).

Overall, I think the paper proposes an interesting unification and generalization of existing state-of-the-art approaches [6, 4], but I think the experimental evaluation needs to be more extensive and clarified to judge the potential significance of the results. The presentation of the paper also needs some polishing as there are many typos and grammatical errors in the current manuscript (see comments below).


####################
*Additional Feedback*

*Positive Highlights*
1. Well motivated anomaly detection approach that unifies existing state-of-the-art deep one-class classification [6] and transformation-based classification [3, 4] approaches that indicates improved detection performance and is applicable to general types of data.
2. The work is well placed in the literature. All relevant and recent related work is included in my view.

*Ideas for Improvement*
3. Extend and clarify the experimental evaluation as discussed in (i) to infer statistical significance of the results.
4. I think many details from the experimental section could be moved to the Appendix leaving space for the additional experiments.
5. Maybe add some additional tabular datasets as presented in [2, 7].
6. Maybe clarify “Classification-based AD” vs. “Self-Supervised AD” a bit more since unfamiliar readers might be confused with supervised classification.
7. Improve the presentation of the paper (fix typos and grammatical errors, improve legibility of plots)
8. Some practical guidance on how to choose hyperparameter $s$ would be good. This may just be a default parameter recommendation and showing that the method is robust to changes in s with a small sensitivity analysis.

*Minor comments*
9. The set difference is denoted with a backslash not a forward slash, e.g. $R^L \setminus X$.
10. citet vs citep typos in the text (e.g. Section 1.1, first paragraph “ ... Sakurada & Yairi (2014); ...”)
11. Section 1.1: “ADGMM introduced by Zong et al. (2018) ...” » “DAGMM introduced by Zong et al. (2018) ...”.
12. Eq. (1): $T(x, \tilde{m})$ in the first denominator as well.
13. Section 2, 4th paragraph: $T(x, \tilde{m}) \in R^L \setminus X_{\tilde{m}}$.
14. $m$, $\tilde{m}$, and $m'$ are used somewhat inconsistently in the text.
15. Section 3: “Note, that it is defined everywhere.”?
16. Section 4: "If $T$ is chosen deterministicaly ..." >> "If $T$ is chosen deterministically ..."
17. Section 5, first sentence: “... to validate the effectiveness our distance-based approach ...” » “... to validate the effectiveness of our distance-based approach ...”.
18. Section 5.1: “We use the same same architecture and parameter choices of Golan & El-Yaniv (2018) ...” » “We use the same architecture and parameter choices as Golan & El-Yaniv (2018) ...”
19. Section 5.2: “Following the evaluation protocol of Zong et al. Zong et al. (2018) ...” » “Following the evaluation protocol of Zong et al. (2018) ...”.
20. Section 5.2: “Thyroid is a small dataset, with a low anomally to normal ratio ...” » “Thyroid is a small dataset, with a low anomaly to normal ratio ...”.
21. Section 5.2, KDDCUP99 paragraph: “Tab. ??” reference error.
22. Section 5.2, KDD-Rev paragraph: “Tab. ??” reference error.


####################
*References*

[1] P. Bergmann, M. Fauser, D. Sattlegger, and C. Steger. Mvtec ad–a comprehensive real-world dataset for unsupervised anomaly detection. In Proceedings of the IEEE Conference on Computer Vision and Pattern Recognition, pages 9592–9600, 2019.
[2] J. Chen, S. Sathe, C. Aggarwal, and D. Turaga. Outlier detection with autoencoder ensembles. In SDM, pages 90–98, 2017.
[3] S. Gidaris, P. Singh, and N. Komodakis. Unsupervised representation learning by predicting image rotations. In ICLR, 2018.
[4] I. Golan and R. El-Yaniv. Deep anomaly detection using geometric transformations. In NIPS, 2018.
[5] X. He, Y. Zhou, Z. Zhou, S. Bai, and X. Bai. Triplet-center loss for multi-view 3d object retrieval. In Proceedings of the IEEE Conference on Computer Vision and Pattern Recognition, pages 1945–1954, 2018.
[6] L. Ruff, R. A. Vandermeulen, N. Görnitz, L. Deecke, S. A. Siddiqui, A. Binder, E. Müller, and M. Kloft. Deep one-class classification. In International Conference on Machine Learning, pages 4393–4402, 2018.
[7] L. Ruff, R. A. Vandermeulen, N. Görnitz, A. Binder, E. Müller, K.-R. Müller, and M. Kloft. Deep semi-supervised anomaly detection. arXiv preprint arXiv:1906.02694, 2019.

**Experience Assessment:**

I have published in this field for several years.

**Review Assessment: Checking Correctness Of Derivations And Theory:**

N/A

**Review Assessment: Checking Correctness Of Experiments:**

I carefully checked the experiments.

**Review Assessment: Thoroughness In Paper Reading:**

I read the paper thoroughly.

---

> ### Author Response · Authors · 2019-11-15
> **Reponse**
>
> We thank the reviewer for the dedicated and mostly positive review. We are pleased the reviewer recognized that our approach is interesting and well motivated and that it convincingly outperforms the state-of-the-art competitors on standard benchmarks.
>
> We sincerely apologize that the editorial quality of the paper was not of the high standard that the reviewer naturally expected. We have significantly revised and improved it, including all the stylistic issues the reviewer highlighted. We believe the quality is now of a high standard.
>
> “consider at least another image dataset besides CIFAR-10, e.g. Fashion-MNIST”: The results for FashionMNIST were added to the paper in the appendix. Overall our method achieves the best performance of all methods.
>
> “On CIFAR-10, do you also consider geometric transformations”: We are using exactly the same geometric transformations as Golan and El-Yaniv [4]. As noted in the paper, geometric transformations are a special case of the affine transformation class. For CNNs to be maximally effective the transformation needs to be locality preserving. Using random affine matrices for images classified by CNNs did not perform competitively as it removed pixel locality information exploited by CNNs. This is different for tabular data where there is no order between different features, making random matrices a good choice of transformation. We updated this insight in the manuscript.
>
> “how do deep networks perform in contrast to the final linear classifier reported on most datasets?”: Results of deep classifiers are significantly better than linear methods for KDD, KDDRev which are data rich (linear accuracy forKDDCup99, KDDRev was around 80%). Results for deep classifiers are roughly similar to linear classifiers for data poor tasks (Thyroid, Arrhythmia).
>
> “ensemble baselines [2] should be considered”: we compared our method an ensemble baseline similar to [2] (implemented by the PyOD package) in exactly the same setting as our experiments. Results can be seen in the appendix. Our method outperforms the ensemble method on the tested datasets. The results are most remarkable on the larger datasets, on which deep classifiers have a distinct advantage.
>
> “Table 2 also seems incomplete with the variances missing”:  Results are copied from Zong et al. did not contain variance, the variance values for these methods are missing in the table. All methods that we ran report variance results. We revised the paper to clarify this.
>
> “How many transformations do you consider on the specific datasets?”: A graph with the accuracies for all datasets is shown in the appendix. Above a certain threshold the number is not critical, the reported experiments used 32 transformations were used for all datasets but Arrhythmia which used 64 (due to its high variance owing to its small size). We present in the appendix results for a larger number of transformations on the smaller datasets (1024 on Arrhythmia and Thyroid) with performance increases on these smaller datasets.
>
> “How is hyperparameter s chosen?”: the value is not very sensitive. We found that a value of s=1.0 performed well in all datasets and is the recommended starting point. Although originally we ran Cifar10 using s=0.1, we present the same experiment with s=1.0 in the appendix with very similar numbers.
>
> We followed the further ideas for improvement proposed by the reviewer. The style and editorial quality of the paper were much improved. The requested experiments were added. We will add further tabular experiments in the final version of the paper. The clarifications requested by the reviewer were added. We also clarified the “Classification” and “Self-supervised” terms.
>
> We are thankful for the reviewer’s detailed comments, which we believe have improved the paper.

---

### Official Review · AnonReviewer2 · 2019-10-21
**Official Blind Review #2**

**Rating:** 8

**Review:**

This paper proposes a novel approach to classification-based anomaly detection for general data. Classification-based anomaly detection uses auxiliary tasks (transformations) to train a model to extract useful features from the data. This approach is well-known in image data, where auxiliary tasks such as classification of rotated or flipped images have been demonstrated to work effectively. The paper generalizes to the task by using the affine transformation y = Wx+b. A novel distance-based classification is also devised to learn the model in such as way that it generalizes to unseen data. This is achieved by modeling the each auxiliary task subspace by a sphere and by using the distance to the center for the calculation of the loss function. The anomaly score then becomes the product of the probabilities that the transformed samples are in their respective subspaces. The paper provides comparison to SOT methods for both Cifar10 and 4 non-image datasets. The proposed method substantially outperforms SOT on all datasets. A section is devoted to explore the benefits of this approach on adversarial attacks using PGD. It is shown that random transformations (implemented with the affine transformation and a random matrix) do increase the robustness of the models by 50%. Another section is devoted to studying the effect of contamination (anomaly data in the training set). The approach is shown to degrade more gracefully than DAGMM on KDDCUP99. Finally, a section studies the effect of the number of tasks on the performance, showing that after a certain number of task (which is probably problem-dependent), the accuracy stabilizes.


PROS:

* A general and novel approach to anomaly detection with SOT results.

* The method allows for any type of classifier to be used. The authors note that deep models perform well on the large datasets (KDDCUP) while shallower models are sufficient for smaller datasets.

* The paper is relatively well written and easy to follow, the math is clearly laid out.


CONS:

* The lack of a pseudo-code algorithm makes it hard to understand and reproduce the method

* Figure 1 (left) has inverted colors (DAGMM should be blue - higher error).
* Figure 1 (right) - it is unclear what the scale of the x-axis is since there is only 1 label. Also the tick marks seem spaced logarithmically, which, if i understand correctly, is wrong.

* The paragraph "Number of operations" should be renamed "Number of tasks" to be consistent. Also the sentence "From 16 ..." should be clarified, as it seems to contrast accuracy and results, which are the same entity. The concept of 'stability of results' is not explained clearly. It would suffice to say: 'From 16 tasks and larger, the accuracy remains stable'.

* In section 6, the paragraph "Generating many tasks" should be named "Number of tasks", to be consistent with the corresponding paragraph in section 5.2. Also the first sentence should be: "As illustrated in Figure 1 (right), increasing the number of tasks does result in improved performance but the trend is not linear and beyond a certain threshold, no improvements are made. And again the concept of 'stability' is somewhat misleading here. The sentence '...it mainly improves the stability of the results' is wrong. The stability is not improved, it is just that the performance trend is stable.

* The study on the number of tasks should be carried on several datasets. Only one dataset is too few to make any claims on the accuracy trends as the number of task is increased.

* The authors should coin an acronym to name their methods.

Overall this paper provides a novel approach to classification-based semi-supervised anomaly detection of general data. The results are very encouraging, beating SOT methods by a good margin on standard benchmarks.


**Experience Assessment:**

I have published one or two papers in this area.

**Review Assessment: Checking Correctness Of Derivations And Theory:**

I assessed the sensibility of the derivations and theory.

**Review Assessment: Checking Correctness Of Experiments:**

I carefully checked the experiments.

**Review Assessment: Thoroughness In Paper Reading:**

I read the paper thoroughly.

---

> ### Author Response · Authors · 2019-11-15
> **Response**
>
> We thank the reviewer for the dedicated and positive review and are pleased the reviewer recognized the novelty of the approach, its state-of-the-art performance and computational scalability.
>
> As requested by the reviewer, pseudo code for the algorithm was added to the paper.
>
> The labels were indeed mislabeled in Fig.1, our approach achieved the better performance. We updated the figure to elucidate all issues brought to our attention by the reviewer.
>
> We run the “number of tasks” experiment on the other datasets, they are shown in the appendix. For all datasets, increasing the number of tasks increases performance up to a certain point. From this point increasing the number of tasks mainly decreases variance between runs. For the smaller datasets, accuracy improves up to a higher number of transformations. We presented the results with the maximal number of transformations in the appendix. We elucidated the text related to this experiment.
>
> We coined the acronym GOAD and use it for our method in the text. Thank you for this helpful suggestion.

---

### Official Review · AnonReviewer1 · 2019-10-23
**Official Blind Review #1**

**Rating:** 8

**Review:**

Review: The paper proposes a technique for anomaly detection. It presents a novel method that unifies the current classification-based approaches to overcome generalization issues and outperforms the state of the art. This work also generalizes to non-image data by extending the transformation functions to include random affine transformations. A lot of important applications of anomaly detection are based on tabular data so this is significant. The “normal” data is divided into M subspaces where there are M different transformations, the idea is to then learn a feature space using triplet loss that learns supervised clusters with low intra-class variation and high inter-class variation. A score is computed (using the probabilities based on the learnt feature space) on the test samples to obtain their degree of anomalousness. The intuition behind this self-supervised approach is that learning to discriminate between many types of geometric transformations applied to normal images can help to learn cues useful for detecting novelties.

Pros:
- There is an exhaustive evaluation and comparison across different types of data with the existing methods along with the SOTA.
- It is interesting to see how random transformations indeed helped to achieve adversarial robustness.
- The method is generalized to work on any type of data with arbitrary number of random tasks. It can even be used in a linear setting if needed for small datasets.

Cons:
- While I liked that an analysis was done to see the robustness of the method on the contaminated data, I would be interested to see a more rigorous comparison in this fully unsupervised setting.


Comments/Question:
Does the selection of the transformation types affect the method performance at all?

In the Results section on Page 7, there are a couple of “??” instead of table numbers.


**Experience Assessment:**

I have published one or two papers in this area.

**Review Assessment: Checking Correctness Of Derivations And Theory:**

N/A

**Review Assessment: Checking Correctness Of Experiments:**

I assessed the sensibility of the experiments.

**Review Assessment: Thoroughness In Paper Reading:**

I read the paper thoroughly.

---

> ### Author Response · Authors · 2019-11-15
> **Reponse**
>
> We thank the reviewer for the dedicated and positive review. We are pleased that the reviewer recognized the novelty and strong performance of our method across many data types, the novel adversarial robustness that it brings and its scalability across different computational regimes.
>
> We presented the contamination results on KDDRev as this was the comparison made in the DA-GMM paper. To address the reviewer’s request for further experiments on contaminated data, we computed the results on the other datasets (Thyroid did not have enough anomalies to perform this experiment). The graphs are presented in the appendix. The trend is similar to that observed in the KDDRev experiment.
>
> The type of transformation affects the results but not very significantly. We present results in the appendix for the affine transformation restricted to (i) permutation and (ii) rotation matrices. The results in line with the full affine transformation (typically a little lower). We have previously experimented with using randomized neural networks as the auxiliary transformations however in our preliminary experiments, the results were not as good as for the linear (affine, rotation, permutation) transformation classes.
>
> We fixed the missing table reference in the revised version of the submission.

---

### Public Comment · ~Thomas_G_Dietterich1 · 2020-04-21
**Some notes and questions for the authors**

We enjoyed reading this paper in my research group. This general paradigm, of training on auxiliary tasks, is an important one, and it is nice to see work building on Golan & El Yaniv's paper. Congratulations!

We have several questions, and we thought asking them publicly would allow you to answer them publicly, which would benefit everyone.

1. How well does the method work as the number of "known" classes is increased? Other researchers, including in our group, have found that many AD and open category methods break down as the number of known classes gets large.

2. In our experiments [1], we have found that the Isolation Forest [2] is an excellent anomaly detector for feature vector data. It is of course vastly cheaper to train than your method. We encourage you to compare against it. Note that you should vary the hyperparameters (number of trees, subsampling size), because as the dimensionality of the data increases, the number of trees should increase.

3. A method that also employs random projections is LODA [3]. It is also extremely efficient. You should compare against it as well.

4. The KDD1999 dataset is extremely simple. Virtually all anomalies can be detected via one-dimensional marginal distributions [4]. It should not be used any more.

5. What is the meaning of the \pm intervals in Tables 1 and 2? Are these 95% confidence intervals on AUC? If so, how were they computed? What source of variation is being controlled for?

6. Why did you report F1 in Table 3 as opposed to AUC? I have never encountered a real anomaly detection application for which F1 is a sensible measure. The usual goal is to detect, say, 99% of anomalies while minimizing false alarms, so Precision @ 99% Detection (Recall) is a good metric. There are some applications where one wants very low false alarms, in which case Recall @ 1% False Alarms is an appropriate metric. You also should provide confidence intervals. You do report $\sigma$: What does this mean and how is it computed? What source of variation is being measured?

7. You did not actually answer Blind Reviewer #3's question about how you set s (although Table 4 is helpful)

8. On p. 3, we could not understand the paragraph that begins "A significant issue with this methodology". Why would a learned classifier only be valid for samples from the training set? What does it mean for the predicted probabilities to have high variance? You aren't using variance as an anomaly score. We were also confused by the following paragraph that says "training P(m|T(x,m)) = 1/M". We think you mean training on a loss function to encourage the predicted distribution to be uniform across the transformations. In general, the notation is non-standard and very confusing.

General Comment:
You don't use the full power of affine transformations (you set b=0), and it isn't clear what benefit the constant offset b would provide. We think it would be more accurate to refer to your technique as using random low-dimensional projections. We suspect that the reason that these work is different from the reason that the geometric transformations of Golan & El Yaniv work. Their transformations require the latent representation to capture some important semantic information about the task, whereas the random projections require the classifier to just preserve random information. For image data, a major challenge is "nuisance novelty" caused by unimportant changes in the background. Such changes are unlikely to be useful for predicting the geometric transformations, whereas intrinsic properties of the objects will be useful.

In contrast, in featurized data, the features are already meaningful. The main issue is sometimes that there are irrelevant features. This is why the sparse random projections of LODA and the random splitting of Isolation Forest are useful. By looking at lower-dimensional interactions, they are less likely to be fooled by global nuisance novelty novelty caused by unique combinations of noisy, irrelevant variables. One way to test this hypothesis might be to use sparse random projections in your method (i.e., where you select a subset of the features and then project them to a space of similar dimensionality). Another experiment would be to add irrelevant features.

References:
[1] A Meta-Analysis of the Anomaly Detection Problem. Emmott, Das, Dietterich, Fern, Wong. arXiv 1503.01158
[2] Liu, F. T., Ting, K. M., & Zhou, Z.-H. (2008). Isolation Forest. 2008 Eighth IEEE International Conference on Data Mining, 413–422. https://doi.org/10.1109/ICDM.2008.17
[3] Pevný, T. (2015). Loda: Lightweight on-line detector of anomalies. Machine Learning, November 2014. https://doi.org/10.1007/s10994-015-5521-0
[4] Siddiqui, M. A., Fern, A., Dietterich, T. G., & Wong, W.-K. (2015). Sequential Feature Explanations for Anomaly Detection. Proceedings of ODD 2015. http://arxiv.org/abs/1503.00038 . See Figure 4

---

> ### Author Response · Authors · 2020-04-26
> **Response**
>
> Thank you for your interest in our work!
>
> The hypothesis that the affine transformations do something similar to LODA is very interesting, however we have some evidence to the contrary. Our method is not only successful when W is a projection matrix (and b=0). It is typically successful when W is a random permutation matrix. It also works in most cases when W is replaced by a random diagonal matrix (so again not projection). Additionally, it works when W=I and a random offset vector b (one can think of it as an additive source separation identification task) - the addition task (W=I, non-zero b) works well with numeric data however it has issues with categorical data (when x is mostly zeros, identifying which transformation b becomes too easy). The trends that we see are typical of RotNet-like methods, the auxiliary task should be not too easy nor too hard. So although averaging over nuisance factors might be contributing a bit, we believe predicting transformations provides the main contribution. We do agree that our method is only as good as the relations present between the variables - so if indeed a bag of words, linear model or a small number of variables are sufficient, classical methods would perform well, whereas if more complex relations need to be learned then deep methods are a good option (cifar10 treated as a tabular dataset is such an example). We also believe that these approaches can benefit from tabular-specific deep architectures over the fully-connected architecture used here, this is a direction for future research.
>
> More detailed response:
>
> In followup research, we run Golan & El-Yaniv when the normal data contains multiple (unlabeled) classes and performance was indeed reduced, so we assume our method will have a similar behavior. We are currently working on improving our method for this setting.
>
> Other baselines: On your suggestion, we evaluated Isolation Forest (IF) and LODA. IF performs very well on the smaller datasets (better than our method), LODA performs comparably to the baseline. This is not too surprising as we suffer from overfitting for small datasets. On KDD and KDDRev (which have more data) - our method significantly outperforms both. We expect that our method will achieve better results for more complex, large datasets.
>
> Protocol - We compared using the protocol in DAGMM, Zong et al., ICLR'18. We agree the protocol has the limitation that you mentioned, however the F1 score is the metric reported by that paper. The source of randomness in the tabular experiments is the split of normal data used for training and testing (and network initialization). The errors are standard deviations over the mentioned number of runs. The source of randomness in image experiments is the network initialization - in line with Golan and El-Yaniv.
>
> Margin - we did not use a principled procedure (or an exhaustive search over the margin values), we tried one or two values. The results did not seem to be very sensitive to them.
>
> Open vs closed set methods - Intuitively, we mean that with closed-set softmax training, we are not able to say in advance that anomalous test data will be classified with high-probability as one of the transformations (correct or incorrect) or have equal probabilities for all transformations - whereas with openset training, anomalous data (with deep representations that are sufficiently different from normal) should have equal probabilities for all transformations.

---

### Decision · Program_Chairs · 2019-12-19

**Decision:**

Accept (Poster)

**Comment:**

The paper presents a method that unifies classification-based approaches for outlier detection and (one-class) anomaly detection. The paper also extends the applicability to non-image data.

In the end, all the reviewers agreed that the paper makes a valuable contribution and I'm happy to recommend acceptance.